# Preliminary Study and Observation of “Kalamata PDO” Extra Virgin Olive Oil, in the Messinia Region, Southwest of Peloponnese (Greece)

**DOI:** 10.3390/foods8120610

**Published:** 2019-11-23

**Authors:** Vasiliki Skiada, Panagiotis Tsarouhas, Theodoros Varzakas

**Affiliations:** 1Department of Food Science and Technology, University of Peloponnese, Antikalamos, 24100 Kalamata, Greece; 2Department of Supply Chain Management (Logistics), International Hellenic University, Kanellopoulou 2, 60100 Katerini, Greece; ptsarouh@mail.teiste.gr

**Keywords:** EVOO, Kalamata PDO, Koroneiki cultivar, Greece, Messinia region, EU regulations, quality and chemical parameters, sterols

## Abstract

While there has been considerable research related to Koroneiki cultivar in different areas in Greece, no systematic work has been carried out on olive oil analysis from one of the most important olive-growing regions in Greece, located southwest of Peloponnese, Messinia. This work is the first systematic attempt to study the profile of Messinian olive oils and evaluate to what extent they comply with the recent EU regulations in order to be classified as “Kalamata Protected Designation of Origin (PDO)”-certified products. Quality indices were measured and detailed analyses of sterols, triterpenic dialcohols, fatty acid composition and wax content were conducted in a total of 71 samples. Messinian olive oils revealed a high-quality profile but, at the same time, results demonstrated major fluctuations from the established EU regulatory limits on their chemical parameters. Results showed low concentrations of total sterols, with 66.7% of the examined samples below the regulated set limits for Kalamata PDO status; high concentrations of campesterol, with a total of 21.7%, exceeding the legal maximum of 4.0%; and a slight tendency of high total erythrodiol content. Fatty acid composition and wax content were within the normal range expected for the extra virgin olive oil (EVOO) category. However, the narrower established PDO limits in specific fatty acids showed some fluctuations in a few cases.

## 1. Introduction

Olive oil is a key element of the Mediterranean diet as well as an exceptional lipid source. Prestigious scientific studies have acknowledged olive oil as a healthy food with multiple utilities in, and benefits for, the human body [1,2]. Nowadays, it is well established that the health-promoting effects of extra virgin olive oil are attributed not only due to its high oleic acid content but also due to its unique bioactive polar phenolic compounds [3,4,5]. As a result, the biological properties, health-promoting effects and nutritive characteristics of extra virgin olive oil have led to a continuous growth in its consumption [6].

Greece is ranked third among olive oil-producing countries, after Spain and Italy, with approximately 16% of the annual production worldwide. Almost 60% of Greece’s arable land is taken up by olive trees. It is the world’s top producer of black olives and has more olive cultivars than any other country worldwide. The annual olive oil production is approximately 300,000–400,000 tons, depending on the harvest year, and 80% of the olive oil produced belongs to the category of extra virgin olive oil (EVOO) [7,8,9]. Hence, olive cultivation in Greece represents not only a crucial resource for rural economies but also an important part of the social, cultural and environmental heritage, as more than 450,000 families work in the fields of olive cultivation [10].

Geographically speaking, almost 70% of olive oil production in Greece is centered in two regions—Peloponnese (39%) and Crete (30%)—with the prefecture of Messinia being the dominant olive-growing area of Peloponnese [10]. Koroneiki cultivar (Olea europeae var. Microcarpa alba) is the indigenous variety in Messinia—the name of which derives from Koroni, a small seaside village southeast of Messinia [11].

Although there are many research publications related to Koroneiki cultivar in different areas in Greece [12,13,14,15,16,17], no systematic work has been carried out on olive oil analysis from the Messinia region. In August 2015, the European Commission approved the extension of the “Kalamata Protected Designation of Origin (PDO) olive oil” from the former province of Kalamata to the rest Regional Unit of Messinia, considerably enlarging the area covered by the PDO [18,19,20,21,22]. On this basis, the new “Kalamata PDO olive oil” introduces more stringent criteria/specifications than those laid down in the European Commission Regulation 2568/1991 for extra virgin oil in order to ensure that the name “Kalamata PDO olive oil” is used only for the area’s olive oil [21,22]. This recent approval, throughout the boundaries of Messinia, could be a very competitive advantage with an important added value, giving a higher market price and a robust commercial presence to “Kalamata PDO Olive oil,” as a PDO trademark is considered an additional guarantee of quality, authenticity, tradition and safety [23,24,25]. However, it is questionable whether Messinian olive oils meet the requirements of the “Kalamata PDO olive oil” profile.

The aim of this study was to investigate, evaluate and report the qualitative and chemical parameters of extra virgin olive oils obtained from the Messinia region. This data will be a useful and important tool in profiling their typical characteristics and evaluating the extent to which they comply with the amended regulation in order to be classified as PDO-certified products. Finally, this study is a motivation for a deeper investigation of the Messinian olive oil, from the southwest region of Peloponnese, which is one of the most important olive-growing regions in Greece and, at the same time, very little investigated.

## 2. Materials and Methods

### 2.1. Geographical Distribution and Selection of Olive Oil Samples

A total of seventy-one (71) olive oil samples were obtained in one successive harvesting year (2014–2015), cultivated in the geographical Messinia region, southwest of Peloponnese, in Greece (see Figure 1). All samples were produced from olive trees representing the typical Koroneiki cultivar of Messinia. Sampling was made from different points in the prefecture of Messinia so as to have the utmost homogeneity. As mentioned earlier, the European Commission recently approved the extension of “Kalamata PDO olive oil” throughout the whole Messinia region, enlarging the area covered by the PDO. On this basis, Messinian extra virgin olive oils may be classified as PDO, if they meet the corresponding parameters [21]. The whole region is characterized by the same climatic conditions as described in the relevant EC Commission Regulation for Kalamata PDO olive oil [19,20,21].

### 2.2. Sampling and Sample Maintenance

Sampling was carried out during the 2014–2015 olive fruit-harvesting period. Provision was made to harvest olive fruits at the optimal stage of maturity. Samples were transferred to local oil mills in solid, vented, food-grade harvest bins or in suitable waist harvest bags. Olive mills were equipped with two or three-phase centrifugal systems (decanters), as olive mills in Messinia operate with both extraction methods (the ratio of two- and three-phase olive mills in Messinia is approximately 50:50). Olive fruits were processed within 24 h, according to the relevant EC Regulation for Kalamata PDO olive oil and the same post-harvest conditions were maintained in all cases. In detail, the leaves were removed from the olive fruits, washed and then sent to the crusher. Malaxation was carried out at low temperatures (27–28 °C) for 30 min according to the above-mentioned regulation. The obtained olive paste was horizontally centrifuged (decanted) (three- or two-phase system) and the resulting olive oil was finally centrifuged. Olive oil samples were stored directly in 1 L air-tight dark-green glass bottles at 4 °C until further analysis. Quality parameters were analyzed in triplicate, while all the other examined chemical parameters were determined in duplicate.

### 2.3. Determination of the Physicochemical Quality Parameters

Free acidity, peroxide value and spectroscopic indices (K_232_ and K_268_) were carried out, following the analytical methods described in Regulation EEC/2568/91 of the European Commission and later amendments [23]. Free acidity was expressed as the percentage of oleic acid and peroxide value was given as milliequivalents of active oxygen per kilogram of oil (meq O_2_ kg^−1^). K_232_ and K_268_ extinction coefficients were calculated from absorption at 232 and 268 nm respectively. Spectrophotometric examination in the ultraviolet provides information on the olive oil quality, its state of preservation and changes brought about by technological processes (due to the presence of conjugated diene and triene systems resulting mainly from oxidation processes). These absorptions are expressed as specific extinctions E (the extinction of 1% *w*/*v* solution of the oil in isooctane, in a 10 mm cell) conventionally indicated by K ‘extinction coefficient’. Free acidity (FA), peroxide value (PV), K_232_ and K_268_ were immediately determined for each sample in order to avoid any kind of olive oil deterioration. Solvents used were purchased from Sigma (St. Louis, MO, USA).

### 2.4. Determination of Sterols and Triterpene Dialcohols

The individual sterols, total sterols and triterpene dialcohols were determined according to the method adopted by EEC/2568/91 regulation, Annexes V with later amendments [23]. The oil sample, with added α-cholestanol (Sigma, St. Louis, MO, USA), as an internal standard, was saponified with potassium hydroxide in ethanolic solution and the unsaponifiable matter was extracted with diethyl ether. The sterol and triterpene dialcohol fractions were separated from the unsaponifiable matter by thin-layer chromatography on a basic silica gel plate (Fluka, Buchs, Switzerland). The fractions recovered from the silica gel were transformed into trimethylsilyl ethers (TMSE) by the addition of pyridine-hexamethyldisilizane-tri-methylchlorosilane (9:3:1, *v*/*v*/*v*) (Supelco, Bellefonte, PA, USA). Sterols (%) and triterpene dialcohol contents were determined with a Shimadzu (GC-2010) gas chromatograph equipped with a flame ionization detector (FID), a DB-5 (30 m × 0.32 mm × 0.25 μm) capillary column and an autosampler injector. The operating conditions were as follows: injection temperature 280 °C, column temperature 265 °C, detector temperature 310 °C, splitting ratio (1:50), flow rate 1.4 mL/min and amount of substance injected 1 μL of TMSE solution. The sterols and triterpene dialcohols were eluted in the following order: cholesterol, 24-methylen-cholesterol, campesterol, campestanol, stigmasterol, Δ7-campesterol, Δ5,23-stigmastadienol, clerosterol, β-sitosterol, sitostanol, Δ5-avenasterol, Δ5,24-stigmastadienol, Δ7-stigmastenol, Δ7-avenasterol, erythrodiol and uvaol (calculated as total erythrodiol). Individual peaks were identified on the basis of their relative retention times with respect to the internal standard. The sum of Δ5,23-stigmastadienol, clerosterol, β-sitosterol, sitostanol, Δ5-avenasterol, and Δ5,24-stigmastadienol represents apparent b-sitosterol. Mean values of duplicate experiments in each sample were used for further statistical analysis.

### 2.5. Determination of Fatty Acid Composition

The fatty acid profile was determined according to the official method of the Regulation EEC/2568/91, Annex IV with amendments [23]. The fatty acid methyl esters (FAME) were obtained by cold alkaline transesterification with methanolic potassium hydroxide solution and extracted with n-heptane. FAME were analyzed on a model GC-2010 Shimadzu chromatograph, equipped with an BPX-70, (60 m × 0.25 mm × 0.25 μm), capillary column and a flame ionization detector (FID). The carrier gas was helium, with a flow of 1.5 mL/min. The temperatures of the injector and detector were set at 250 and 260 °C respectively and the oven temperature was increased gradually from 165 to 225 °C in 35 min. The injection volume was 1 μL. Quantification was achieved using a FAME standard mixture purchased from Sigma (St. Louis, MO, USA). The results were expressed as a percentage of individual fatty acids. Analytical-grade methanol, heptane, and potassium hydroxide were purchased from Sigma (St. Louis, MO, USA).

### 2.6. Determination of Wax Content

The wax content of olive oil samples was determined according to the Regulation EEC/2568/91, Annex IV with later amendments [23]. A suitable amount of internal standard (lauryl arachidate) was added to 0.5 g of olive oil sample and then fractionized by chromatography on a hydrated silica gel column. The chromatographic elution was carried out with a mixture of n-hexane/diethyl ether, keeping a rate of flow of approximately 15 drops every 10 s. The subsequent fraction was completely dried and finally resolved in 2 mL of n-hexane. Waxes were analyzed on a model GC-2010 Shimadzu chromatograph equipped with an on-column injector, a flame ionization detector and a MEGA-5 HD (10 m × 0.32 × 0.10 mm) capillary column. The operating conditions were as follows: detector temperature 370 °C; the column temperature was increased from 80 to 160 °C at 40 °C/min and up to 340 °C at 5 °C/min for 7 min; the amount of substance injected was 1 μL of the n-hexane solution. The identification of the peaks was based on retention time by comparison with wax mixtures of known retention times analyzed under the same conditions.

### 2.7. Statistical Analysis

Results were expressed as the mean values ± standard deviation (SD). Data were processed with MINITAB 18 software. Thus, it is possible to extract the minimum and the maximum value of the sample, mean, and standard deviation (SD). Differences between means were tested for statistical significance using analysis of variance (ANOVA). The statistical significance level was set at *p* < 0.05. Moreover, Principal Component Analysis (PCA) was applied to study the relations between the extraction method (two- or three-phase decanter) on the examined chemical properties.

## 3. Results and Discussion

### 3.1. Qualitative Parameter of Messinian Olive Oils, Greece

As shown in Figure 2, of the 71 virgin olive oils analyzed, all are classified as extra virgin olive oil (EVOO), as far as the qualitative indices are concerned, according to the European Regulation (EEC) 2568/91 as amended, with the exception of two samples in total, which were not within the accepted acidity value of 0.80% and excluded for further analysis. The content of free acids is an important quality factor, extensively used as the major criterion for the classification of olive oil at various commercial grades.

According to the relevant Commission Regulation for Kalamata PDO olive oil, stricter quality specifications compared to EU Regulation 2568/91 have been laid down. [20,21]. As shown in Figure 2, a high percentage of the examined Messinian olive oil samples (88.73%) did not exceed the threshold of 0.50% in acidity, which is defined as the upper limit for Kalamata PDO olive oils. The mean acidity value was 0.34% and ranged from 0.17 to 0.76 (Table 1). In addition, peroxide value and spectrophotometric analysis, crucial indices of olive oil oxidation, were within and quite below the upper limit established by EC Regulation for the EVOO category as presented in Figure 2. In particular, peroxide value for the tested samples ranged from 3.64 to 11.96 meq. O_2_ kg^−1^, with a mean value at 7.24 meq. O_2_ kg^−1^ (Table 1). Likewise, K_232_ and K_268_ values had a mean value of 1.55 and 0.13, respectively, with only one sample surpassing the Kalamata PDO limit for K_268_ value.

It should be noted that as peroxide value is a quality indicator of the primary products of auto-oxidation (hydroperoxides) of an olive oil, poor post-extraction conditions (e.g., inappropriate storage and packaging) may result to a fast increase in peroxide value above the defined limit of 14 meq O_2_·kg^−1^, excluding olive oils from the PDO labeling.

In general, the above observations depict the highest quality of Messinian olive oil production, one of the most important olive-growing regions in Greece, but most importantly highlight how crucial it is to retain those qualitative characteristics, especially with the recent approval of the European commission to expand “Kalamata PDO olive oil” in the whole regional unit of Messinia (21).

### 3.2. Analysis of Sterolic Profile and Triterpenic Dialcohol Content of Messinian Olive Oils, Greece

Phytosterols are important components of the unsaponifiable fraction of olive oil beneficial for the human health and nutrition. Sterol composition and content are broadly used for the control of olive oil authenticity and adulteration. Sterol content varies between 1000 and 3000 mg/kg depending the botanical variety, olive ripening, storage conditions and geographical origin [26,27,28,29,30,31]. Numerous studies have shown that each variety has a characteristic sterol “fingerprint”. Therefore, those minor components can be considered as an important and useful tool for detecting oil adulteration and/or classifying virgin olive oils in accordance with their variety [32,33,34,35]. The influence of geographical origin on the sterol composition of virgin olive oil has been evaluated by various authors, pointing out the great potential of different analytical techniques followed by chemometric tools for this purpose [36,37,38].

Although several studies have been conducted for cv Koroneiki in other regions of Greece, mainly in Crete [39,40,41,42,43], very little information is available in the literature regarding the sterolic profile of cv Koroneiki in Peloponnese generally and more precisely in the Messinia region.

In the present study, we evaluated the sterolic composition of the examined Messinian olive oils. Table 2 lists the mean values expressed as percentages of the total sterols and their standard deviations of the main sterols present in the olive oil sampled. The main sterols detected were β–sitosterol, Δ5-avenasterol and campesterol, with mean values of 80.73%, 12.28% and 3.71%, respectively. The first two represent over 90% of the total sterol content, with β-sitosterol being the most abundant phytosterol (over 80% of the total sterol content). The calculated parameter, the apparent β-sitosterol, falls within the established regulatory limits, with a mean value of 94.63%. Finally, the cholesterol and Δ7-stigmastenol values were low and quite below the limits set by EU regulation (0.5%), with a mean value of 0.11% and 0.19% of total sterols, respectively (Table 2).

In contrast, several major deviations were observed in the case of the sterolic profile for the Messinian olive oils. Most importantly, 43.5% of the examined olive oil samples did not surpass the required limit of 1000 mg/kg in total sterol concentration according to the EEC Regulation 2568/91. In addition, as illustrated in Figure 3, the regulated limit for Kalamata PDO olive oil is established at 1100 mg/kg. As a result, a really high percentage (66.7%) of the examined samples was below the established PDO limit. The mean total sterols content was 1033 mg/kg and ranged from 744 to 1283 mg/kg. A similar case was observed in campesterol, where a total of 21.7% of the examined samples exceeded the legal maximum of 4%, with a mean value of 3.71% and ranged from 2.78 to 4.70%. A trend of higher campesterol has also been reported for cv Koroneiki, as well as for other cultivars cultivated in different countries [44,45], whereas the total sterol concentration of the most studied Spanish and Italian cultivars is always within the minimum limit of 1000 mg/kg [46,47,48,49,50].

Although no information exists in the literature regarding Kalamata PDO olive oils, results show that cv Koroneiki in the Messinian region shows a clear tendency of low concentrations of total sterols and high concentrations of campesterol. Low mean values on total sterol concentration for cv Koroneiki were reported earlier in Crete, in 2001, by Stefanoudaki et al., who studied the effect of drought stress on olive oil characteristics, without giving emphasis on the mentioned tendency [41].

In general, such problems (fluctuations from EU regulations) could inevitably raise questions regarding the authenticity of Kalamata PDO extra virgin olive oils in the olive oil sector, and so they certainly require further investigation.

It is known that total erythrodiol levels are high in solvent-extracted oils, indicating adulteration with olive-pomace oil [51]. The mean total erythrodiol content was 2.85% (Table 2). However, a small but noteworthy percentage of 8.06% of the examined samples exceeded the upper set limit of 4.5% as shown in Figure 3. A possible assumption may be the inappropriate higher degree of olive crushing during the extraction process, leading to an increase in erythordiol levels from the olive’s exocarp.

Finally, almost no significant differences were observed in the sterol composition and triterpene dialcohols using the two industrial decanters (*p* > 0.05) (please see Appendix A). The amount of water added during oil extraction does not affect their levels due to their lipophilic nature and because they are sparingly soluble in water. This is in agreement with previous reported data for cv Koroneiki among other cultivars [40,52].

### 3.3. Fatty Acid Composition of Messinian Olive Oil, Greece

A crucial parameter for the quality and characterization of olive oil is the fatty acid composition [53]. In the present study, thirteen fatty acids were identified. As shown in Table 3, the variability of fatty acid composition was within the normal range expected for the EVOO category in all the examined samples. The mean values for the major fatty acids were 76.70% for oleic acid (C18:1), 12.02% for palmitic acid (C16:0), 6.09% for linoleic acid (C18:2), 2.53% for stearic acid (C18:0), and 0.92% for palmitoleic acid (C16:1). The percentage of the monounsaturated oleic acid ranged from 70.67% to 81.40% and depicts the beneficial health impact of Messinian olive oils and the competitive profile of Kalamata PDO olive oils to the olive oil market. Palmitic acid, the second most abundant fatty acid ranged from 9.54% to 13.56%, the poly-unsaturated linoleic acid ranged between 4.2% and 12.01%, stearic ranged from 1.98% to 3.12% and palmitoleic acid ranged from 0.64% to 1.43%.

Despite the fact that all samples met the standards for the EVOO category as mentioned above, according to the EEC regulation 2568/91, we should stress that quite narrower limits exist for specific fatty acids such as palmitoleic, stearic, oleic and linoleic acid, according to the relevant EU regulation for Kalamata PDO olive oil as shown in Table 3 [20,21]. This led to fluctuations in a number of cases (10/69) from the legal PDO limits. Hence, further investigation should be carried out.

Finally, our results showed that there were only minor changes in fatty acid composition when different decanters were used. In particular, as shown in Table 3, the only significant differences were observed in C17:0, C17:1, C18:0 and C20:0 which are fatty acids of lower proportions in the total fatty acid percentage. This is in agreement with several other studies, where it is reported that there is very little effect of the extraction method on fatty acid composition [40,54,55].

### 3.4. Analysis of Wax Content of Messinian Olive Oil, Greece

Waxes are important constituents of olive oil used in order to distinguish olive oil obtained by pressing and that obtained by extraction (olive-residue oil) [56,57]. Waxes are present on the external fruit wax cuticle in olives so as to protect the fruit from transpiration and insect damage [57]. In dry hot weather, plants produce more waxes in order to control the rate of transpiration so that reduction of water loss is achieved. As a result, high temperature increases the wax production as a mechanism of fruit defense from environmental factors (climate) [58]. Generally, it has been found that wax compositions are influenced mainly by cultivar, harvest year and malaxation conditions [59,60,61].

In the present study, six wax esters were detected in the GC chromatogram: C36, C38, C40, C42, C44 and C46. Due to the latest amendment of the EEC regulation [23], the sum of wax esters was classified in two groups. Since only C42, C44 and C46 are now included in the most recent EU regulation for extra virgin olive oil, they were grouped together as Wax Esters (WEs 42–46). Since the sum of C40, C42, C44 and C46 was previously calculated, they were also grouped together as Total Wax Esters (TWEs 40–46). As shown in Table 4, the mean value of WEs 42–46 of the examined olive oil samples was 28.38 mg/kg and ranged from 16.89 to 58.33 mg/kg, well short of the upper legal limit of 150 mg/kg. Respectively, the TWEs 40–46 had a mean value of 67.20 mg/kg and ranged between 42.84 to 140.31 mg/kg. Although there is no previous study on wax determination for Kalamata PDO olive oils, according to our knowledge, the obtained results were similar to the wax content of olive oils extracted in other hot climates such as those in southern Italy for cv Carolea and in Australia for cv Koroneiki [44.46]. Finally, we found no differences in wax content caused by extraction (*p* > 0.05) (please see Appendix A).

Using chemometric analysis, we verified that the extraction method (two- or three-phase decanter) causes non or minor changes on the examined chemical characteristics.

The score plot of Principal Component Analysis (PCA) is used to assess the data structure and detect clusters, outliers, and trends. As shown in Figure 4 groupings of data on the plot based on two-phase and three-phase decanters in Messinian olive oil samples showed that the points are randomly distributed around zero. As a result, no correlations for both two-phase and three-phase samples were presented. Therefore, the extraction method (two- or three-phase decanter) has verified no changes to the chemical parameters examined.

## 4. Conclusions

The evaluation of extra virgin olive oils produced in the Messinia region, southwest of Peloponnese, denoted some challengeable characteristics. On the one hand, the results depict the high qualitative profile of Messinian olive oils, which is in agreement with similar studies examining cv Koroneiki from different geographical regions of Greece. On the other hand, major fluctuations were observed from the established EU regulatory limits. Most importantly, results show that Messinian extra virgin olive oils show low concentration of total sterols, with 66.7% of the examined samples being below the regulated set limits for Kalamata PDO status. Although no information exists in the literature regarding Kalamata PDO olive oils, as mentioned previously, analysis of VOOs from cv Koroneiki in completely different geographical regions such as Crete and Australia, has also shown a tendency of low total sterol concentration. Thus, low mean value in total sterols may clearly depict a “special characteristics” for Koroneiki cultivar, yet completely opposed to the existing standard limits of Kalamata PDO status. In contrast, total sterol concentration of the most studied Spanish and Italian cultivars is always quite above the limit of 1000 mg/kg.

In addition, our results show that olive oil samples of cv Koroneiki in the Messinia region present a high concentration of campesterol, with a total of 21.7%, exceeding the legal maximum of 4.0% and a light tendency of high total erythrodiol content. A trend of higher campesterol has been reported for cv Koroneiki, cv Barnea and cv Cornicabra cultivated in other geographical regions such as in Australia and Spain. Furthermore, although the fatty acid composition of the examined samples was within the range for the EVOO category, the extremely narrow established PDO limits in specific fatty acid composition may result in further fluctuations excluding Messinian olive oils from PDO certification. As far as wax content is concerned, although no information exists in the literature for Messinian olive oils for comparison, the obtained results are within the regulatory EU limits. Finally, in accordance with previous reported data for cv Koroneiki in Greece, the extraction method (two- or three-phase decanter) caused non or minor changes on the examined chemical characteristic.

In general, as PDO-certified products are a crucial strategic tool to enhance rural economy and development, through the added value of the PDO trademark, in terms of the higher price such products can enjoy, the above-mentioned deviations could inevitably lead to a controversy regarding the authenticity of Kalamata PDO extra virgin olive oils in the olive oil sector and consequently result in diminishing its reputation.

As this work is the first systematic attempt focusing on the evaluation of “Kalamata PDO olive oil” characteristics, further in depth research, with a higher number of samples and more crop years, is under way. The continued study of Messinian olive oils with the addition of more examined parameters (e.g., sensory analysis) will provide adequate datasets and allow supporting the improvement of the current EU regulation through the update and the re-adjustment of the established limits for Kalamata PDO status.

## Figures and Tables

**Figure 1 foods-08-00610-f001:**
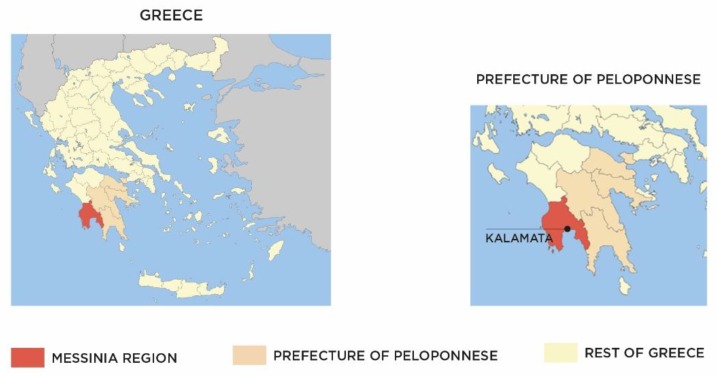
Map of Greece, focusing on the Messinia region (in red), southwest of the Prefecture of Peloponnese (in orange). Adapted from Wikipedia [22].

**Figure 2 foods-08-00610-f002:**
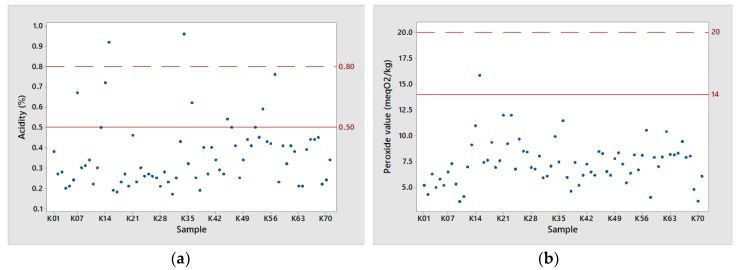
(**a**) Scatter plots visualizing the qualitative parameters (**a**) acidity, (**b**) peroxide value (**c**) K_232_ extinction coefficient and (**d**) K_268_ extinction coefficient, respectively, of the 71 examined Messinian olive oil samples numbered as K1–K71, *N* = 71. Dotted line: limits according to EEC/2568/91 for extra virgin olive oil (EVOO) category; straight line: limits according to Council Regulation (EC) 510/2006 for Kalamata Protected Designation of Origin (PDO) olive oil [20,21,23].

**Figure 3 foods-08-00610-f003:**
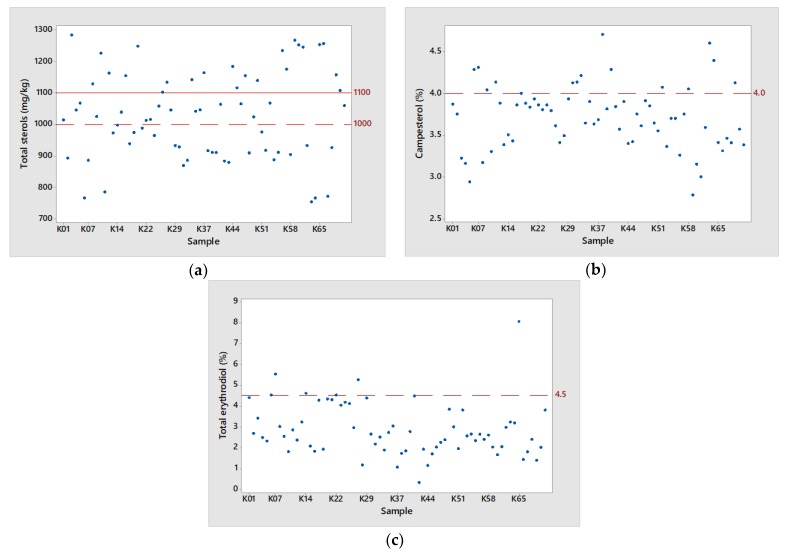
(**a**) Scatter plots visualizing the chemical parameters: (**a**) total sterols (43.5% of the examined olive oil samples did not surpass the EEC limit of 1000 mg/kg and 66.3% of the examined olive oil samples did not surpass the PDO limit of 1100 mg/kg in total sterols); (**b**) campesterol (21.7% of the examined olive oil samples exceeded the legal maximum of 4%); (**c**) total erythrodiol (8.06% of the examined olive oil samples exceeded the upper set limit of 4.5%). Note. Dotted line: limits according to EEC/2568/91 for the EVOO category; straight line: limits according to Council Regulation (EC) 510/2006 for Kalamata PDO olive oil [20,21,23].

**Figure 4 foods-08-00610-f004:**
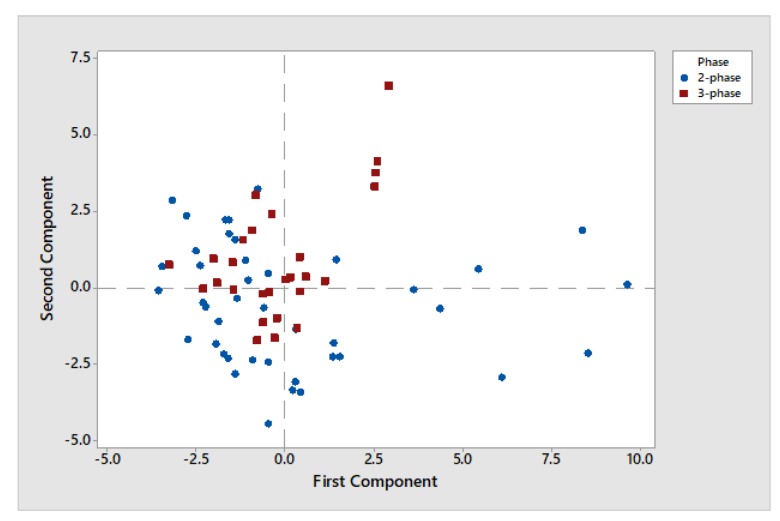
Score plot of Principal Component Analysis (PCA) for two-phase and three-phase decanters in Messinian olive oil.

**Table 1 foods-08-00610-t001:** Qualitative parameters of Messinian olive oils.

Parameter	Mean ± SD	Min–Max	EEC Limit for the EVOO Category	PDO Limit
Free acidity (%)	0.34 ± 0.13	0.17–0.76	≤0.80	≤0.50
Peroxide value (meqO_2_ kg^−1^)	7.24 ± 1.88	3.64–11.96	≤20	≤14
K_232_	1.55 ± 0.14	1.33–2.14	≤2.50	≤2.20
K_268_	0.13 ± 0.01	0.08–0.21	≤0.22	≤0.20

Results are expressed as the means ± standard deviation (SD). *N* = 69. EEC = European Commission, EVOO = Extra Virgin Olive Oil, PDO = Protected Designation of Origin.

**Table 2 foods-08-00610-t002:** Sterolic profile and triterpene diols determined in Messinian olive oil, Greece.

Sterols and Triterpene Diols	Mean ± SD	EEC Limit	PDO Limit
Cholesterol (%)	0.11 ± 0.03	≤0.5	≤0.5
24-methylene-cholesterol%	0.32 ± 0.09		
Campesterol%	3.71 ± 0.38	≤4.0	≤4.0
Campestanol%	0.05 ± 0.03	<campesterol	<campesterol
Stigmasterol%	0.74 ± 0.19		
Chlerosterol%	0.85 ± 0.07		
β-Sitosterol%	80.73 ± 3.73		
Sitostanol%	0.37 ± 0.30		
Δ-5-avenasterol%	12.28 ± 3.96		
Δ-5,24-stigm/dienol%	0.29± 0.10		
Δ-7-stigmastenol%	0.19 ± 0.09	≤0.5	≤0.5
Δ-7-avenasterol%	0.28 ± 0.11		
Apparent b-Sitosterol%	94.63 ± 1.07	≥93.0	≥93.0
Total erythrodiol%	2.85 ± 1.25	≤4.5	≤4.5
Total sterols (mg/kg)	1033.3 ± 150.1	≥1000	>1100

Results are expressed as the means ± standard deviation (SD). *Ν* = 69.

**Table 3 foods-08-00610-t003:** Percentage composition (%) of major fatty acids in Messinian olive oils and influence of the extraction method on the fatty acid profile.

Fatty Acid (%)	Mean ± SD	Min–Max	EEC Limit	PDO Limit	Two-Phase	Three-Phase	Difference *p*-Value
					Mean ± SD	Mean ± SD	
Myristic C14:0	0.01 ± 0.00	0.00–0.02	≤0.03		0.01 ± 0.00	0.01 ± 0.00	n.s
Palmitic C16:0	12.02 ± 0.74	9.54–13.56	7.50–20.00	10.0–15.0	11.96 ± 0.81	12.11 ± 0.61	n.s
Palmitoleic C16:1	0.92 ± 0.13	0.64–1.43	0.30–3.50	0.6–1.2	0.93 ± 0.13	0.89 ± 0.08	n.s
Heptadecanoic C17:0	0.05 ± 0.02	0.03–0.15	≤0.40		0.04 ± 0.01	0.06 ± 0.03	0.003
Heptadecenoic C17:1	0.08 ± 0.04	0.06–0.24	≤0.60		0.07 ± 0.01	0.10 ± 0.05	0.016
Stearic C18:0	2.53 ± 0.19	1.98–3.12	0.50–5.00	2.0–4.0	2.44 ± 0.16	2.67 ± 0.16	0.00
Oleic C18:1	76.70 ± 1.96	70.67–81.40	55.00–83.00	70–80	76.92 ± 2.19	76.36 ± 1.52	n.s
Linoleic C18:2	6.09 ± 1.60	4.20–12.01	2.50–21.00	4.0–11.0	6.05 ± 1.91	6.14 ± 0.95	n.s
Linolenic C18:3	0.68 ± 0.07	0.51–0.86	≤1.00		0.66 ± 0.08	0.69 ± 0.06	n.s
Arachidic C20:0	0.44 ± 0.03	0.33–0.50	≤0.60		0.43 ± 0.04	0.45 ± 0.02	0.012
Eicosenoic C20:1	0.31 ± 0.02	0.27–0.35	≤0.50		0.31 ± 0.02	0.30 ± 0.02	n.s
Behenic C22:0	0.14 ± 0.01	0.09–0.17	≤0.20		0.14 ± 0.01	0.14 ± 0.01	n.s
Lignoceric C24:0	0.05 ± 0.00	0.034–0.08	≤0.20		0.05 ± 0.00	0.05 ± 0.00	n.s

Results are expressed as the mean ± standard deviation (SD). n.s = non-significant. Differences between means of two-phase vs. three-phase centrifugal systems were tested for statistical significance using analysis of variance (ANOVA). The statistical significance level was set at *p* < 0.05.

**Table 4 foods-08-00610-t004:** Wax esters of Kalamata PDO olive oils produced in Messinia (southwest of Peloponesse).

Parameter (mg/kg)	Mean ± SD	Min–Max	EEC Limit	PDO Limit
Wax Esters C40–C46 (WEs)	67.20 ± 18.88	42.84–140.31	≤250	≤250
Wax Esters C42–C46 (TWEs)	28.38 ± 9.62	16.89–58.33	≤150	≤150

Results are expressed as the means ± standard deviation (SD).

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
