# Peer review of "Preliminary Study and Observation of “Kalamata PDO” Extra Virgin Olive Oil, in the Messinia Region, Southwest of Peloponnese (Greece)"

_foods, 2019, doi:10.3390/foods8120610_

Round 1
Reviewer 1 Report
I have no further comments.
Author Response
ok thank you.
Reviewer 2 Report
The Authors have corrected the manuscript accordingly with my comments from their previous submission. Though the quality of this resubmitted work has increase, there are some minor issues that need to be solved before final acceptance.
The quality of figures is low. I.e., the DPI of Figure 1 is not satisfactory. Figures 2 and 3, the y-axis labels are unreadable due to the too small font size and low resolution. Figure 4 should be moved to the middle of the page.
Author Response
Thank you for your comments. We have upgraded all figures. The DPI of Fig. 1 is very high. Figures 2 and 3 have been corrected on the y axis incorporating all units and with a higher font size.
Please find attached.

This manuscript is a resubmission of an earlier submission. The following is a list of the peer review reports and author responses from that submission.
Round 1
Reviewer 1 Report
In the submitted work the Authors performed an attempt to study the profile of Messinian olive oils and evaluate to what extent they comply with the recent EU regulations in order to be classified as “Kalamata PDO” certified products. A lot of experimental work, yet rather routine, has been done and described properly, though the statistical analysis is limited.
Lines 35-36, this statement needs a reference. The same applies to the sentence starting in line 42.
There are numerous editorials mistakes, such as lack of space (line 42, before "Hence), lack of a dot (lines 44 and 80) or extra space at lines 55 and 211, etc. Please look carefully through the whole manuscript to remove them.
Lines 84-85: this looks bad. Instead of citing the whole hyperlink to the wikipedia website, please insert the proper citation at the end of manuscript.
Lines 157-158: "The SD is defined as the square root of the variance, and 157 is often used in place of the variance to describe the distribution spread." This sentence is trivial and therefore it should be removed.
K232 and K268 - those parameters should be explained.
The statistical analysis is very limited. Since the Authors collected a lot of data (c.a. 30 parameters for each of 71 samples) it would be nice if they could perform kind of a chemometric (PCA) analysis to study if there are some relations between the features of the studied samples.
Table 3, the p-value for staeric acid was really 0.00 or is it a mistake?
Author Response
Please find attached our replies to the reviewers' questions.

Reviewer 2 Report
Synopsis of Results: The present work proposes to be the fist systematic study of “Kalamata PDO olive oil” produced in Messinia region, southwest of Peloponnese. Results revealed the high quality of the Messinian oil, but also the fluctuation from the established EU regulatory limits on their chemical parameters.
No systematic work has been carried out on olive oil analysis from the region of Messinia. Research is, thus, an original contribution to the existing literature, but the data provided are insufficient. One of the main limitations of this study is the low number of samples analysed as underlined also by the authors. Sampling was carried out only during the 2014-2015 harvesting period: to make more complete the statistical analysis, additional data should be presented.
It would have been useful to compare the obtained data with other cultivars from the same region, as well as the same cultivar from other regions (if there are data available in the literature).
Minor points:
The introduction could illustrate the important beneficial effects of secondary metabolites in PDO EVOOs (i.e. Foods 2017, 6(10), 90; https://doi.org/10.3390/foods6100090; Antioxidants 2019, 8(7), 217; https://doi.org/10.3390/antiox8070217)
Author Response
Οne of the main limitations of this study is the low number of samples analysed as underlined also by the authors. Sampling was carried out only during the 2014-2015 harvesting period: to make more complete the statistical analysis, additional data should be presented.
The number of samples taken was seventy one (71) olive oil samples and were obtained in one successive harvesting year (2014-2015). We did all these analysis also and believe it was a good start.The statistical analysis was carried out effectively and we also added a PCA plot to make the data more concrete.
Many authors refer to one harvesting year and we have mentioned it in our paper. The reason why we did not compare the next year was because in Messinia region the quality was not good due to weather conditions, hence sampling was not adequate. The following year was even worse due to problems with an insect.
It would have been useful to compare the obtained data with other cultivars from the same region, as well as the same cultivar from other regions (if there are data available in the literature).
In “Results and Discussion” we discussed and compared our obtained results with literature data of the same cultivar (cvKoroneiki) in other regions (please see for example lines 234-243, 254-255, 284-288, 311-315). However, in order to be more comparative, we also included a paragraph in conclusion to compare our results with data obtained in other geographical regions for the same cultivar (Koroneiki).
Similar studies examining cv Koroneiki in different geographical regions of Greece [14-17, 39-43].
Although no information exists in the literature regarding Kalamata PDO olive oils, as mentioned previously, analysis of VOOs from cv Koroneiki in completely different geographical regions such as Crete and Australia, has also shown a tendency of low total sterols concentration [41,44]. Thus, low mean value in total sterols may clearly depict a “special characteristics” for Koroneiki cultivar, yet completely opposed to the existing standard limits of Kalamata PDO status. In contrast, total sterols concentration of the most studied Spanish and Italian cultivars is always quite above the limit of 1000 mg/kg [46-50].
In addition, our results show that olive oil samples of cv Koroneiki in Messinia region present high concentration in campesterol with a total of 21.7% exceeding the legal maximum of 4.0% and a light tendency of high total erythrodiol content. A trend of higher campesterol has been reported for cv Koroneiki, cv Barnea and cv Cornicabra cultivated in other geographical regions such as in Australia and Spain [44-45].
As far as wax content is concerned, although no information exists in the literature for Messinian olive oils for comparison, the obtained results are within the regulatory EU limits. Finally, in accordance with previous reported data for cv Koroneiki in Greece [40], the extraction method (2 or 3-phase decanter) caused non or minor changes on the examined chemical characteristic.
Note: In the geographical boundaries of Messinia, Koroneiki is the predominant olive variety. So comparison with other cultivars in the same region would not be so feasible.
The introduction could illustrate the important beneficial effects of secondary metabolites in PDO EVOOs (i.e. Foods 2017, 6(10), 90; https://doi.org/10.3390/foods6100090; Antioxidants 2019, 8(7), 217; https://doi.org/10.3390/antiox8070217).
Done; In the introduction of the revised version, we included a sentence highlighting the beneficial effects of phenolic compounds, with relevant references [3-5]